# Large Enhancement in Thermal Conductivity of Solvent−Cast Expanded Graphite/Polyetherimide Composites

**DOI:** 10.3390/nano12111877

**Published:** 2022-05-30

**Authors:** Fatema Tarannum, Swapneel S. Danayat, Avinash Nayal, Rajmohan Muthaiah, Roshan Sameer Annam, Jivtesh Garg

**Affiliations:** School of Aerospace and Mechanical Engineering, University of Oklahoma, Norman, OK 73019, USA; fatema-1@ou.edu (F.T.); sdanayat@ou.edu (S.S.D.); nayal1@ou.edu (A.N.); rajumenr@ou.edu (R.M.); anna0003@ou.edu (R.S.A.)

**Keywords:** thermal conductivity, expanded graphite, porous, effective medium model

## Abstract

We demonstrate in this work that expanded graphite (EG) can lead to a very large enhancement in thermal conductivity of polyetherimide−graphene and epoxy−graphene nanocomposites prepared via solvent casting technique. A *k* value of 6.6 W⋅m^−1^⋅K^−1^ is achieved for 10 wt% composition sample, representing an enhancement of ~2770% over pristine polyetherimide (*k*~0.23 W⋅m^−1^⋅K^−1^). This extraordinary enhancement in thermal conductivity is shown to be due to a network of continuous graphene sheets over long−length scales, resulting in low thermal contact resistance at bends/turns due to the graphene sheets being covalently bonded at such junctions. Solvent casting offers the advantage of preserving the porous structure of expanded graphite in the composite, resulting in the above highly thermally conductive interpenetrating network of graphene and polymer. Solvent casting also does not break down the expanded graphite particles due to minimal forces involved, allowing for efficient heat transfer over long−length scales, further enhancing overall composite thermal conductivity. Comparisons with a recently introduced effective medium model show a very high value of predicted particle–particle interfacial conductance, providing evidence for efficient interfacial thermal transport in expanded graphite composites. Field emission environmental scanning electron microscopy (FE−ESEM) is used to provide a detailed understanding of the interpenetrating graphene−polymer structure in the expanded graphite composite. These results open up novel avenues for achieving high thermal conductivity polymer composites.

## 1. Introduction

Increased thermal diss0ipation in modern electronic devices has led to a demand for thermally conductive materials with superior thermal conductivity (*k*) [1,2]. The light weight, high corrosion resistance, and excellent processability of polymeric materials make them attractive for thermal management applications [3,4]. However, poor thermal conductivity (0.5 W⋅m^−1^⋅K^−1^) [5,6] of polymers limits their application in the efficient heat removal process. The addition of high thermal conductivity fillers such as graphene or carbon nanotubes has been shown to significantly enhance the thermal conductivity of polymer graphene nanocomposites [7,8,9,10,11,12,13]. Different effects have been investigated to enhance the effectiveness of graphene in enhancing k, such as alignment and percolation effects. Alignment of carbon nanomaterials [14,15,16,17,18,19,20] takes advantage of their thermal conductivity in one direction (in−plane for graphene nanoplatelets and along the axis for carbon nanotubes). However, the enhancement achieved in *k* through alignment effects is anisotropic, potentially limiting the application of such composites. One of the most promising approaches has been the percolative effect, where graphene–graphene contact is used to bypass the polymer, leading to significantly higher thermal conductivity enhancement [8,13,21]. More recently, expanded graphite has been shown to yield very high composite *k* values [22,23,24,25,26,27]. In this paper, we explore the use of expanded graphite (EG) for enhancing the thermal conductivity of polyetherimide (PEID) due to the unique applications of this polymer in electrical systems [28,29,30], which can benefit from efficient heat dissipation. In particular, we show that expanded graphite/polymer composites prepared through solvent casting can lead to more efficient heat transfer due to an almost continuous network of graphitic sheets over long−length scales, which overcomes the problem of low interfacial thermal conductance at graphene–graphene contact in typical percolation environments. We have measured a composite thermal conductivity of 6.6 W⋅m^−1^⋅K^−1^ at just 10 wt% filler concentration. This value represents the highest *k* value measured in a polymer composite at this low weight fraction of filler. Presented results provide new avenues for achieving efficient thermal management in a wide range of applications.

Intercalation of acid molecules and oxidizing agents into graphite followed by rapid heating (900 °C) causes intercalated agents to gasify, producing enough pressure to push the adjacent graphite layers apart, resulting in expanded graphite (Figure 1a). Figure 1b shows the graphite intercalated compounds (GICs), while Figure 1c shows expanded graphite with a worm−like structure. EG achieved through such expansion has a highly porous (Figure 1d,e), lightweight structure with a very low density (0.002–0.02 g⋅cm^−3^) and exhibits high mechanical strength (10 MPa), thermal conductivity (25–470 W⋅m^−1^⋅K^−1^), and electrical conductivity (10^6^–10^8^ S⋅cm^−1^) [31]. As a result, EG has emerged as a promising material with applications such as flame retardancy [32], phase−change material [33,34], electrodes [35,36], electrochemical sensors [37], fuel cells [38,39], batteries [40,41] and supercapacitors [42,43].

Significant research has been performed into the use of EG in enhancing the thermal conductivity of polymer graphene nanocomposites. Wu et al. [44] measured the thermal conductivity of individual expanded graphite particles using a T−type method and reported a value ~335 W⋅m^−1^⋅K^−1^ for EG particles. Tao et al. [45] prepared EG/PDMS composite using a hot−press technique and reported a high thermal conductivity value of 4.7 W⋅m^−1^⋅K^−1^ at 10 wt% EG composition. Zhao et al. [46] measured a high thermal conductivity of 3.5 W⋅m^−1^⋅K^−1^ in EG/paraffin wax composites at 25 wt% composition. Song et al. [47] measured a thermal conductivity of 1.66 W⋅m^−1^⋅K^−1^ at 20 wt% composition in EG/MgCl_2_·6H_2_O composite. Wei et al. [48] created a network of expanded graphite particles using stearic acid and polyethylene wax and measured a high thermal conductivity of 19.6 W⋅m^−1^⋅K^−1^ using ~25 vol% expanded graphite. Che et al. [49] employed synergistic effects between expanded graphite and carbon nanotubes to achieve a high thermal conductivity of ~3.0 W⋅m^−1^⋅K^−1^ at ~20 wt% composition in high−density polyethylene (HDPE) composites.

In this work, we show that expanded graphite can lead to a remarkable increase in the thermal conductivity of polyetherimide composites prepared via the approach of solvent casting. Expanded graphite has a highly porous structure with interconnected graphitic walls, as seen in the SEM image of expanded graphite in the air in Figure 1e. Figure 1f shows the separation of adjacent graphite layers, which occurs due to the release of gases during the expansion procedure. Such a porous structure is absent in graphene nanoplatelets, also used in this work, and presented in Figure 1g. The graphitic sheets form a continuous network over long−length scales, allowing efficient conduction of heat. The approach of solvent casting offers the advantage of preserving this porous structure of expanded graphite as seen in the field emission environmental scanning electron microscopy (FE−ESEM) images of expanded graphite while it is embedded in the polymer composite, in Figure 1h–j. The comparison of Figure 1e (showing EG in the air) and Figure 1h–j (showing EG in the composite) clearly show that the porous structure of EG is retained within the composite. This is due to the use of solvent casting, which does not exert large forces on expanded graphite, preserving its porous structure, unlike microcompounding, where large shear forces distort this structure [48,50]. Such a porous structure gets infused with the dissolved polymer during the casting process resulting in a highly thermally conductive interpenetrating network of graphene and polymer.

The resulting interpenetrating polymer graphene structure from the solvent casting method allows for very efficient heat transfer. This can be seen by observing that graphitic nanosheets in this network are continuous for very long−length scales (the lateral dimensions of these expanded graphite particles can range from a few hundred to thousands of microns), as shown in Figure 1j (arrows indicate heat flow over long continuous paths). The high in−plane thermal conductivity (*k*~2000 W⋅m^−1^⋅K^−1^) [12] of graphene nanosheets forming this network allows very efficient heat transfer over long−length scales, resulting in high composite thermal conductivity. We later show through X−ray diffraction and Raman analysis that the graphitic layers forming the walls of this network retain the chemical structure and exact interlayer spacing of graphite and can thus be expected to have the high thermal conductivity values reported for graphite nanoplatelets (*k_in_*~2000 W⋅m^−1^⋅K^−1^, and *k_out_*~10 W⋅m^−1^⋅K^−1^) [12].

A unique advantage of this continuous network can be understood by noticing that even at bends/turns, the graphitic sheets are still continuous, allowing for a very high thermal conductance (low interfacial thermal resistance) at such junctions. The above effect of low interfacial resistance at bends/turns in an expanded graphite network offers an advantage over the effect of percolation [8,13,21,49,51,52,53]. Percolation involves enhancing heat transfer through establishing discrete particle–particle contact resulting in efficient heat conduction along a network of graphene particles. While this enhances thermal conduction by bypassing the low thermal conductivity polymer, the interfacial thermal resistance at the contact between discrete graphene particles in percolative networks can still be significant. Konatham et al. [54] performed molecular dynamics simulations to show that thermal contact resistance at graphene–graphene contact is 5.5 × 10^−8^ K/W, which is comparable to graphene−polymer interfacial thermal resistance [8]. In expanded graphite composites, however, the continuous nature of graphene sheets at bends/turns results in much more efficient interfacial thermal transport at junctions, resulting in a superior thermal conductivity enhancement overall. Another advantage of expanded graphite−mediated *k* enhancement over percolation effect is that typically very high volume concentrations are required to achieve particle–particle contact for percolation. In the case of solvent−cast expanded graphite/polymer composites, however, the continuous graphitic networks are present at all volume fractions, enabling achieving high *k* even at low particle concentrations.

The polymers chosen for this work, namely polyetherimide and epoxy, enable the advantage of superior thermal interaction with graphene [55,56]. Both polyetherimide and epoxy have oxygen groups in their molecular structure that can enable strong thermal interaction through hydrogen bonding with the oxygen groups [57,58,59] in expanded graphene. Such oxygen groups are introduced in expanded graphite during intercalation (with an oxidizing agent) and the expansion process. Evidence for the presence of oxygen groups in expanded graphite is provided through X−ray photoelectron spectroscopy (XPS) [60,61,62].

We further investigated the effect of sonication parameters on thermal conductivity enhancement and showed that lower sonication time and power result in larger expanded graphite particles in the composite, which allows for heat to be conducted efficiently along longer lengths scales, resulting in higher thermal conductivities. Finally, we also compare the measured thermal conductivity results with a recently introduced effective medium model that takes graphene–graphene contact into account for thermal conductivity prediction.

## 2. Experimental Details

### 2.1. Materials

Graphite intercalated compound (GIC) or expandable graphite (EPG), with an average particle size of ≥180 μm (+100 mesh size: ~92%) and expansion ratio 290:1 (ASB−3570), were bought from a graphite store [63]. Graphene Nanoplatelets AO−4 (60 nm thickness and lateral size 7 μm) were acquired from a graphene supermarket [64]. The epoxy resin used for the study was EPIKOTE RESIN MGS RIMR 135, and the hardener used was EPIKURE CURING AGENT MGS RIMH 137, both purchased from Hexion [65]. Commercially available N, N−dimethylacetamide (DMAc), and polyetherimide (PEID) with a melt index of 18 g/10 min (337 °C/6.6 kg) and a density of 1.27 g/mL were obtained from Alfa Aesar [66] and Sigma Aldrich [67]. The organic solvents N−N, Dimethylformamide (DMF), and Acetone were purchased from the University of Oklahoma chemical stock room.

### 2.2. Fabrication of the EG/Polymer Composites

The fabrication procedure of the EG/PEID composite is illustrated in Figure 2a and the chemical structure of PEID is presented in Figure 2b. For EG/PEID composite preparation, expandable graphite was first placed into a furnace at 900 °C for approximately 30–60 s in a crucible to obtain the required quantity of EG filler. EG so obtained was then dispersed into 20 mL DMAc. Separately PEID pellets were dissolved into 50 mL DMAc at 130 °C for 1 h. The DMAc solution with EG was mixed with a polymer solution and blended for 3 h at 130 °C, followed by a short−time (~40 s) probe sonication at 20% amplitude using probe−sonicator (VCX 750 Sonicator, PRO Scientific Inc., Oxford, Connecticut, USA; power output: 750 W, frequency: 20 kHz). The EG blended with a polymer solution of ~25–30 mL was cast into a petri dish. Lastly, the petri dish was kept at 100 °C for 24–48 h to produce the composite film. Likewise, 2.5 wt%, 7.5 wt%, and 10 wt% EG/PEID composite films were prepared using this same procedure. For comparison, the graphene nanoplatelet (GnP)/PEID composite films were also prepared using the same solution casting technique for graphene with 60 nm thickness, respectively.

To prepare epoxy/expanded graphite composites, resin was added to 90 mL N,N−dimethylformamide (DMF) solution and stirred while heating at 150 °C to obtain a homogeneous mixture. Expanded graphite was added to this solution and stirred for one hour. The solution was then tip sonicated for 40 s followed by stirring at 150 °C until the solvent completely evaporated. After the solvent evaporated, a thick mixture of EG/epoxy was obtained. This mixture was then spread over a PTFE sheet and kept in a vacuum oven at 140 °C for 15 h to remove any residual solvent present in the mixture. On cooling the mixture to room temperature, a hardener was added to it and mixed to obtain a homogeneous viscous paste. This paste was then transferred to aluminum molds and cured at 90 °C for 20 h.

## 3. Characterization

Thermal Conductivity (*k*): *k* of EG/polymer composites was measured by the laser flash technique. A Netzsch LFA 467 Hyperflash (Netzsch, Selb, Germany) laser was used to measure the through−thickness thermal diffusivity of the samples. A total of 8–12 samples of 12.5 mm diameter and 0.3–0.4 mm thickness were used to measure the thermal diffusivity (α) at room temperature (23 °C). The samples were coated with graphite spray before the measurement to efficiently absorb heat from a flash lamp, and an average of 6–8 measurements was reported. This laser flash technique induces heat by a laser pulse on one surface of the sample, and the temperature rise is captured on the other surface of the sample as a function of time. α is determined by LFA using the following equation: α = (0.1388 d^2^)/t_1/2_, where α is the thermal diffusivity (mm^2^/s), t_1/2_ is the time to obtain half of the maximum temperature on the rear surface, and d denotes the sample thickness (mm). The thermal conductivity was calculated using *k* = α×ρ×Cp, where *k*, *ρ*, and Cp represent the thermal conductivity, density, and specific heat constant of the sample, respectively. In this work, the density and specific heat of the composite samples were calculated using a gas pycnometer (AccuPyc II 1340, Micromeritics Instrument Corporation, Norcross, GA, USA) and differential scanning calorimetry (DSC) (DSC 204F1 Phoenix, Netzsch, Selb, Germany).

*Scanning Electron Microscopy (SEM):* Morphological characterization of EG filler and EG/polymer composites was carried out by high−resolution field emission environmental scanning microscopy (Quattro S FE−ESEM, Thermofisher Scientific, Waltham, MA, USA). This SEM was operated in secondary electron (SE) mode at an accelerating voltage of 20 kV. To prepare the samples for SEM imaging, liquid nitrogen was used to crack down the composite to image over the cross−sectional area. Montage large area mapping of EG fillers has been captured using MAPS software of FE−ESEM.

*Raman Spectroscopy (RS):* Raman spectroscopy (RS) was performed using a DXR3 SmartRaman Spectrometer (Thermofisher Scientific, Waltham, MA, USA) to collect the data over the range from 3250 to 250 cm^−1^, laser wavelength λ_L_ = 633 nm, spectral resolution = 0.16 cm^−1^, and imaging resolution = 702 nm for the EG and GIC samples. An Olympus BX 41 microscope with 5× objective, 10 s exposure time for 15 accumulations, and 3 scans per sample were used to collect the spectra.

*X−ray Diffraction (XRD):* A PANalytical Empyrean Diffractometer (Malvern Panalytical Ltd., Grovewood Rd, Malvern WR14 1XZ, UK) produced the information regarding the crystal structure of EG compared to GIC using Bragg−Brentano focusing geometry at room temperature. A total of 3 kW Cu Kα radiation (λ = 1.5406 Å) with a scan range of 2θ = 5 to 80°and step size of 0.013°.

*X−ray Photoelectron Spectroscopy (XPS):* X−ray Photoelectron Spectroscopy (XPS) was performed for GIC and EG samples by Thermo Scientific K−alpha XPS (Thermofisher Scientific, Waltham, MA, USA). A Kα gun source was used to excite the sample, and the data were collected for an acquisition time of ~70 s at 400 µm spot size. The passing energy of 200 eV was used to find the carbon (C), oxygen(O), and sulfur (S) peak in this analysis spectrum. The atomic percentage of C, O, and S were investigated using the Avantage software. To determine the atomic percentage, this software was used to perform the curve fitting in accordance with Gaussian and Lorentzian functions.

## 4. Result and Discussion

### 4.1. Thermal Conductivity Results

The measured thermal conductivities of EG/PEID and EG/epoxy composites are shown in Figure 3a,b. Thermal conductivity of solvent−cast EG/PEID composites is measured to be around 6.6 W⋅m^−1^⋅K^−1^ at 10 wt% filler composition. This value represents a remarkable 2770% enhancement over the *k* (0.23 W⋅m^−1^⋅K^−1^) of pristine PEID, providing new avenues for high thermal conductivity composites. Similarly, the measured thermal conductivity of EG/epoxy composites also shows remarkable enhancement at very low loadings of EG. At just 2 wt% EG composition, a thermal conductivity value of 1.74 W⋅m^−1^⋅K^−1^ is achieved for the epoxy composite, representing an enhancement of 1025% over pristine epoxy (0.16 W⋅m^−1^⋅K^−1^). Table 1 and Figure 3e show that the measured value is significantly higher than similar graphene/polymer composites either based on (a) uniformly dispersed graphene, (b) other expanded graphite−based methods, and (c) graphene−based percolative networks.

We first compare the measured values against previous results reported for the percolation effect based on establishing a graphene–graphene network. A key advantage of expanded graphite is that a large enhancement in *k* value is achieved even at low graphene loading, as opposed to the case of percolation, where typically much higher particle concentrations are required to achieve particle–particle contact. Kargar et al. achieved around 6 W⋅m^−1^⋅K^−1^ in graphene/epoxy composites at a high graphene loading of 35 vol% through the percolation effect [21]. Percolation was also found to yield a thermal conductivity of ~5.5 W⋅m^−1^⋅K^−1^ in Boron–Nitride/epoxy composites [21] at a volume loading of 45 vol%. A high *k* of 6.6 W⋅m^−1^⋅K^−1^ is achieved in this work through the use of just 10 wt% EG content. Even in percolative *k* enhancement, the thermal interfacial resistance at particle–particle contact can be relatively high. Konatham et al. reported a thermal boundary resistance of 5.5 × 10^−8^ K/W, almost as high as at the graphene−polymer contact [54]. Expanded graphite, achieved through the solvent casting approach, leads to continuous graphite networks (Figure 4a–d), overcoming the issue of low particle–particle interfacial thermal conductance in percolative environments. Significant enhancement in *k* value of EG/PEID composite has been achieved due to such continuous graphitic network of EG as presented in Figure 4b,d throughout the composite film.

The high *k* achieved in this work is also due to the use of solvent casting, which offers the advantage of preserving the porous structure of expanded graphite within the composite (seen in Figure 1f,i,j). This is due to only moderate forces being exerted on expanded graphite during the solvent casting approach. Figure 1f,i compare the porous structure of expanded graphite before and after it is embedded into the composite and show that the porous structure of EG is largely retained within the composite. This is in contrast to microcompounding (melt−blending), where large shear forces during the compounding process can exfoliate expanded graphite, converting it into a nanoplatelet−like morphology. Mokhtari et al. [68] discussed such exfoliation of expanded graphite through microcompounding.

We also compare measured *k* values of EG/PEID composite with those of GnP/PEID composites in Figure 3a,b. The measured thermal conductivity of EG/PEID composites is dramatically higher relative to that achieved using graphite nanoplatelets. At 10 wt% composition, the *k* of EG/PEID composite (6.6 W⋅m^−1^⋅K^−1^) is higher by 572% relative to GnP/PEID composite (1.0 W⋅m^−1^⋅K^−1^). Similarly, at 2 wt% composition, *k* of EG/epoxy composite (1.74 W⋅m^−1^⋅K^−1^)) is higher by ~600% relative to GnP/epoxy composite (0.25 W⋅m^−1^⋅K^−1^). At the low graphene content used in this work, GnPs are well separated by polymer; high interfacial thermal resistance between GnPs and polymer then restricts the enhancement achievable through the use of graphene nanoplatelets. These results highlight the large advantage of expanded graphite over graphene nanoplatelets in enhancing composite thermal conductivity.

We further investigated the effect of sonication parameters used during composite preparation on *k* enhancement. Figure 3c,d show that lower sonication time and amplitude lead to the higher thermal conductivity of EG/PEID composite. To understand the effect of sonication parameters on the structural integrity of EG, we performed an FE−ESEM analysis. As the porous interpenetrating network has a beneficial impact on thermal conductivity enhancement (shown in Figure 4a–d), high−resolution images are obtained to visualize the effect of sonication time on this porous network structure (Figure 5a,b). The image of 10 wt% EG/PEID composite (Figure 5a,b) prepared with 3 min sonication time at 20% amplitude shows an absence of porous structure within the composite, suggesting that longer sonication time causes the porous structure to be damaged. FE−SEM images of 10 wt% GnP/PEID composite are also presented in Figure 5c,d.

While the above images show the expanded graphite embedded in the polymer, we also show expanded graphite before it is embedded into the polymer in Figure 5e,f. These images show that fragile, porous graphite is broken into nanosheets after 3 min sonication time, whereas 40 s sonication time has a negligible effect on EG filler structure. Short−time sonication thus offers the advantage of preserving the structure of expanded graphite while also allowing uniform dispersion into the polymer matrix.

In the next section, we compare measured *k* values of EG/PEID composite with theoretical predictions based on a recently introduced effective medium model by Su et al. [69]. Comparison with theoretical predictions highlights the advantage of expanded graphite and provides evidence for the outlined mechanism of heat conduction along continuous graphitic paths.

**Table 1 nanomaterials-12-01877-t001:** Comparison of k_⊥_ for different polymer graphene and EG−polymer composite.

Filler	Matrix	Fraction	$k_{⊥}$ $\left(W\cdot m^{-1}\cdot K^{-1}\right)$	$k_{matrix}$ $\left(W\cdot m^{-1}\cdot K^{-1}\right)$	Enhancement (%)	Preparation Method	Ref.
GnPs/MWCNT	PS	(3.5/1.5) vol%	1.02	0.19	437	Melt mixing + Synergistic effect	[70]
Graphite	PP	40 wt%	5.4	0.25	2060	Compression molding	[71]
Multilayer GnP	Epoxy	10 vol%	5.1	0.21	2300	Solvent casting, higher sheer mixing	[8]
fGO	Epoxy	5 wt%	0.21	0.16	34	Solution casting	[72]
EG	LLDP	5.78 wt%	1.85	0.33	461	Melt mixing, 3D network formation	[48]
EG	PMMA	7 wt%	0.47	0.125	276	Water-assistedmelt extrusion	[73]
EG	LDPE	10 wt%	0.5	0.32	56	Melt mixing	[74]
EG	PDMS	10 wt%	4.7	0.18	2511	Solvent casting, hot press	[45]
EG	PEG	10 wt%	1.324	0.298	344	Melt mixing	[75]
EG	Paraffin	25 wt%	3.16	0.18	1695	Melt mixing	[46]
EG	PEID	30 wt%	1.6	0.2	700	Solvent mixing, melt mixing followed by injection molding	[76]
EG	PEID	10 wt%	6.6	0.23	2770	Solution casting	This work

PS—polystyrene; PP—polypropylene; PMMA—Poly(methyl methacrylate); LDPE—low−density polyethylene; PDMS—polydimethylsiloxane; PEG—Polyethylene glycol.

### 4.2. Effective Medium Model for Thermal Conductivity Prediction

The effective medium model introduced by Su et al. [69] includes the effect of both graphene−polymer and graphene–graphene thermal contact resistance. Effective composite thermal conductivity $k_{e}\;$through this model is computed by solving the following equation:

(1)$$c_{0}\frac{k_{0}-k_{e}\;}{k_{e}+\left(k_{0}-k_{e}\right)/3}+\frac{c_{1}}{3}[\frac{2\left(k_{11}-k_{e}\right)}{k_{e}+S_{11}\left(k_{0}-k_{e}\right)}\;+\frac{\left(k_{33}-k_{e}\right)}{k_{e}+S_{33}\left(k_{3}-k_{e}\right)}]=0$$
where c_0_ and $c_{1}\;$are the concentrations of the embedding matrix and filler material, respectively, $k_{11}$ and $k_{33}$ are the effective in−plane and through−plane thermal conductivities of graphitic nanosheets. The effective thermal conductivities take into account the effect of interfacial thermal resistance. $S_{11}\;$and $S_{33}$ are the shape parameters related to the aspect ratio of graphitic nanosheets, given by the following equations.
$$S_{11}=S_{22}=\frac{\alpha }{2{\left(1-\alpha^{2}\right)}^{3/2}}\left[\arccos\alpha -\;\alpha {\left(1-\alpha^{2}\right)}^{1/2}\right],\;\alpha \;<\;1$$
$$S_{33}=1-S_{11}$$

In the above equations, α is the aspect ratio (thickness/lateral dimension) of the graphitic nanosheets, and *k_0_* is the thermal conductivity of an interlayer surrounding graphene sheets. The role of this interlayer is to include the effect of interfacial thermal resistance at graphene−polymer and graphene–graphene contacts.

The effective thermal conductivities $k_{11}$ and $k_{33}$ are computed using the in−plane and through−plane thermal conductivities of graphene ($k_{1}$ and $k_{3},$ respectively) and the interlayer properties through the following equations,
(2)$$k_{11}=k_{0}\left[1+\frac{\left(1-c_{int}\right)\left(k_{1}-k_{0}\right)}{c_{int}S_{11}\left(k_{1}-k_{0}\right)+k_{0}}\right]$$
(3)$$k_{33}=k_{0}\left[1+\frac{\left(1-c_{int}\right)\left(k_{3}-k_{0}\right)}{c_{int}S_{11}\left(k_{3}-k_{0}\right)+k_{0}}\right]$$

In the above equations, $c_{int}$ is the concentration of the interlayer. The values of different parameters used in this effective medium model are described below.

In the Table 2, graphene thickness and length are the thickness and length of the graphitic sheets forming the walls of the interpenetrating network. Average values of these parameters were obtained from microscopy to be 10 μm for lateral length and 10 nm for thickness. The thermal conductivity of the polymer phase is taken to be 0.23 W⋅m^−1^⋅K^−1^ from our measurements (in suitable agreement with the literature). The in—plane ($k_{1}$) and through—plane thermal conductivities ($k_{3}$) of graphene were taken to be 2000 W⋅m^−1^⋅K^−1^ and 10 W⋅m^−1^⋅K^−1^, respectively [12,77].

The interfacial resistance between graphene and polymer is modeled as an interlayer in the above theory. The thickness of this interlayer was nominally taken to be 2 nm, and its thermal conductivity was assumed to be 0.04 W⋅m^−1^⋅K^−1^, resulting in an interfacial resistance between graphene and polymer of 5 × 10^−8^ m^2^K/W [8], a well−accepted value for interfacial thermal resistance between graphene and polymer.

A key parameter in the above model is the interfacial resistance at graphene–graphene contact. This value was determined by fitting the measured values to the effective medium model. A suitable agreement between measured thermal conductivities and predicted values was obtained by using a graphene–graphene interlayer thermal conductivity of 0.17 W⋅m^−1^⋅K^−1^ (see Figure 3a and Figure 6), which is more than four times higher than the interlayer thermal conductivity of polymer graphene (0.04 W⋅m^−1^⋅K^−1^). This interlayer thermal conductivity of 0.17 W⋅m^−1^⋅K^−1^ at graphene–graphene contact corresponds to a graphene–graphene interfacial thermal resistance of 1.17 × 10^−8^ m^2^K/W. This value is significantly smaller than the interfacial resistance predicted for regular graphene–graphene contact [54] (5 × 10^−8^ m^2^K/W), providing evidence that graphene sheets are in superior contact (covalently bonded) at bends/turns in an expanded graphite network, compared to the contact between discrete graphene particles. The analysis points to the advantage of the continuous graphitic network achieved through the use of expanded graphite via solvent casting technique in enhancing the thermal conductivity of polymer composites. We further show the effect of lower graphene–graphene contact conductance by decreasing the graphene–graphene contact thermal conductivity in the model. It is seen that as the graphene–graphene contact thermal conductivity is decreased from 0.17 to 0.10, the effective composite thermal conductivity is decreased from 6.6 W⋅m^−1^⋅K^−1^ to 4.4 W⋅m^−1^⋅K^−1^ (by almost 33.3%), indicating the importance of superior graphene–graphene contact conductance for achieving overall high composite *k* values. We next discuss the characterization of EG filler and EG/PEID composites.

### 4.3. Morphologies of EG Filler and EG/PEID Composite

Analysis through FE–ESEM reveals the structural integrity of EG structure before and after the preparation of polymer composite. SEM (Scanning electron microscopy) images enable understanding of morphological differences between graphene nanoplatelets (GnPs), Graphene Intercalation compound (GIC), and thermally expanded graphite (EG) (Figure 1a). Thermal expansion of GIC with an average diameter of ~180 μm and thickness of 1–150 μm turns it into a worm–like structure. Uneven expansion resulting from the expansion of the intercalation compound at 900 °C leads to the separation of expandable graphite into multiple layers resulting in a porous network with an average edge size of 10–20 μm. These pores allow the formation of the interpenetrating graphene–polymer network where the pores are wetted with PEID polymer. While Figure 1f shows expanded graphite before it is used to prepare composite, FE–SEM also is used to study the structure of EG even within the polymer composite. EG structures in EG/PEID composites with 7.5 and 10 wt% filler are presented in Figure 4a–d. These figures show that solvent casting clearly preserves the porous network structure of expanded graphite, enabling the creation of the highly thermally conductive interpenetrating graphene–polymer network within the composite.

### 4.4. Analysis of Crystal Structure by XRD and Raman Spectroscopy

X-ray diffraction (XRD) analysis was performed to determine the crystal structure and interlayer spacing of GIC and EG. Figure 7a shows a strong diffraction peak at 2θ = 26.133° (002) for GIC, slightly shifted from the case of natural graphite 2θ = 26.5° [78,79,80,81]. A weaker peak (004) is observed at 2θ = 54.37° [82] for GIC. The small shift in peak for GIC is attributed to the presence of intercalated compounds. On the contrary, a reduced sharp peak is visible at 2θ = 26.35° (002) [83] for EG (the inset of Figure 7a), closer to the (002) graphitic carbon structure. A clear diminution is observed in the intensity at (002) peak, which is caused due to disorder in graphitic morphology [84] after the expansion process. Still, a mostly aligned peak position in EG indicates the existence of intact chemical structure of graphite and interlayer order [85,86].This interconnected and stacked structure of EG enables better thermal transport throughout the polymer composite [25]

Nondestructive Raman analysis was also performed to further analyze the crystal structure before and after the thermal expansion. Raman spectroscopy of Figure 7b exhibits two inherent peaks of G band and D band at ~1580 cm^−1^ and ~1350 cm^−1^ for graphitic material [87]. The G band signifies the stretching of defect−free sp^2^ carbon of hexagonal ring due to in−plane tangential stretching of the carbon−carbon bonds [88], and the D band represents the vibrational mode caused by the amorphous disordered structure of sp^3^ hybridized carbon [60]. The 2D band also can be seen at around 2700 cm^−1^ and represents a second−order two−photon process [89]. Raman spectra of GIC clearly show those characteristic peaks (G, D, and 2D bands). In contrast, G and 2D bands are present, but the presence of the D band is negligible in EG Raman spectra. I_D_/I_G_ ratio is typically used to characterize the defective state of graphene. The absence of a D band in EG suggests the presence of a highly ordered defect−free graphite structure in EG. This high degree of the ordered structure of graphite, even after thermal expansion, has a strong beneficial impact on the thermal conductivity enhancement of polymer graphene composite as it preserves the intrinsic high thermal conductivity of graphene itself.

### 4.5. XPS Analysis of GIC and EG Filler

XPS analysis was further performed to investigate the concentration of carbon (C), oxygen (O), and sulfur (S) elements before and after the thermal expansion of GIC, as presented in Table 3. Figure 7c shows that two peaks of C1s and O1s at ~285 eV and ~532 eV [67] are present for GIC and EG filler, but the S2p peak (~169 eV) [90] is only visible in GIC spectra because of the included intercalated compounds. The atomic percentage of carbon increases from 85.14% to 95.76%, and oxygen reduces from 13.16% to 4.24% after expansion. This is attributed to the fact that thermal expansion at 900 °C releases oxygen contents. The presence of small amounts of oxygen groups in EG allows favorable interactions with oxygen groups in PEID through hydrogen bonding, leading to efficient interfacial thermal transport. This further enhances the thermal conductivity of EG/PEID composite.

## 5. Conclusions

In summary, we demonstrate that expanded graphite (EG) can lead to a large enhancement in the thermal conductivity of EG/PEI composites prepared through solvent casting. At 10 wt% EG composition, a high thermal conductivity of 6.6 W⋅m^−1^⋅K^−1^ is measured, representing an enhancement of 2770% over pristine polyetherimide. This large enhancement in thermal conductivity is found to be due to a network of continuous graphene sheets over long−length scales achieved through the solvent casting technique, which preserves the interconnected porous structure of expanded graphite within the composite. Even at bends/turns in graphene sheets in such a network, the sheets are covalently bonded, which minimizes the interfacial thermal resistance at junctions, enhancing heat transfer. Overall, the resulting structure allows highly efficient heat conduction over long−length scales along the continuous graphitic sheets. This results in the observed high thermal conductivity of the composite through the use of EG. The thermal conductivity of EG/PEID composite is also found to dramatically exceed that of graphene nanoplatelet (GnP)/PEID composites by 572% for 10 wt% filler composition. At low filler loading, GnPs are well separated by polymer and the resulting high graphene−polymer interfacial thermal resistance results in low effective GnP/PEID thermal conductivity. Presented results highlight the advantage of expanded graphite in enhancing the thermal conductivity of polymer composites and can lead to novel avenues for achieving efficient thermal management in a wide array of technologies.

## Figures and Tables

**Figure 1 nanomaterials-12-01877-f001:**
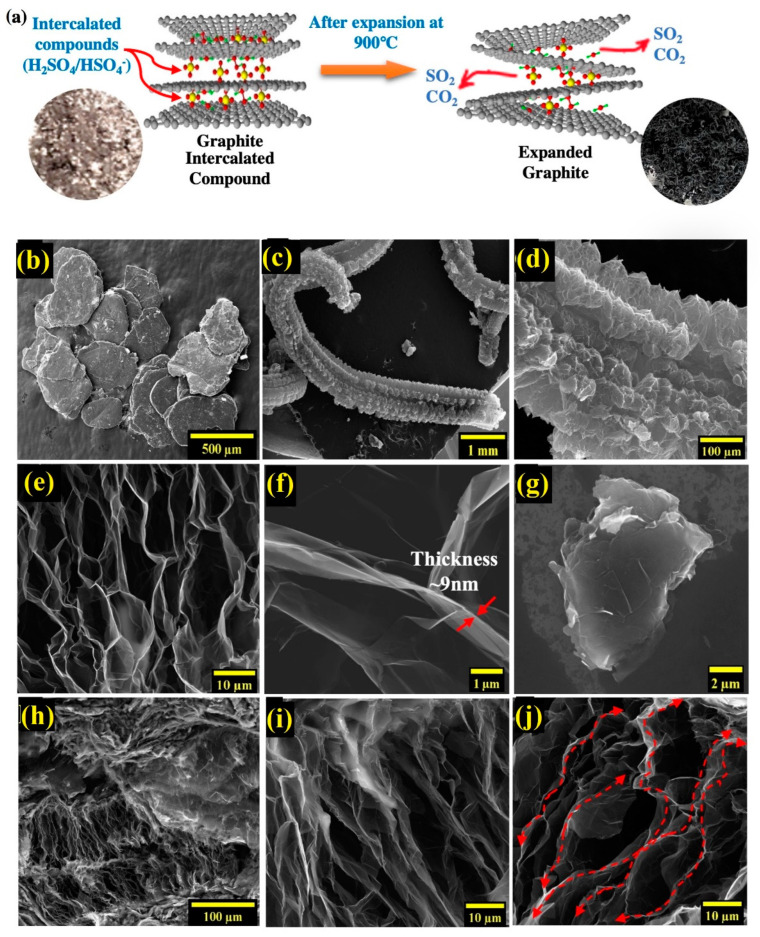
(**a**) Schematic illustration of expanded graphite from graphite intercalated compounds through thermal expansion. SEM image of (**b**) GIC, (**c**) expanded graphite after thermal expansion. High−resolution FE−ESEM images of EG at magnification of (**d**) 100×, (**e**) 1500×, (**f**) 12,000×, (**g**) FE−ESEM images of 60 nm graphene nanoplatelet. FE−ESEM images of EG/PEID composite at (**h**) 350×, (**i**,**j**) 3500×.

**Figure 2 nanomaterials-12-01877-f002:**
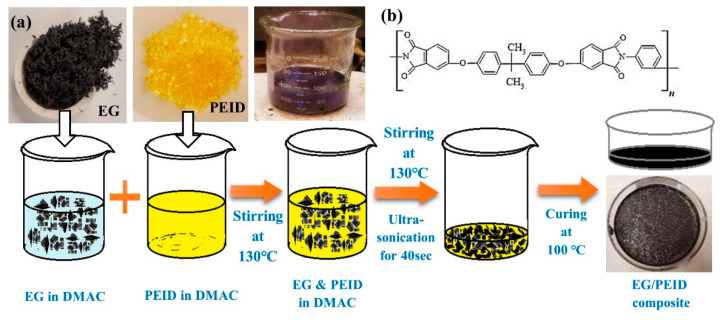
(**a**) Schematic for the preparation process of EG/PEID composites, (**b**) chemical structure of PEID.

**Figure 3 nanomaterials-12-01877-f003:**
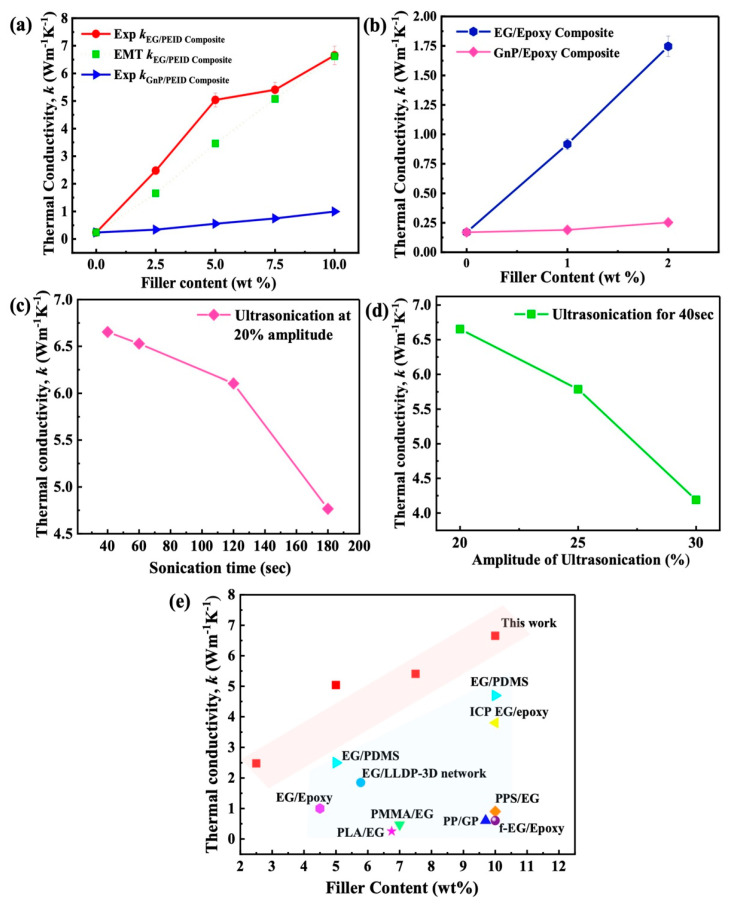
(**a**) Thermal conductivity value of EG/PEID (experimental and effective medium theory (EMT)) and GnP/PEID composite with different filler content (2.5, 5, 7.5, and 10 wt%), (**b**) thermal conductivity value of EG/epoxy composite with different filler content (0.5, 1, 1.5, and 2 wt%); (**c**) *k* value of EG/PEID composite with different sonication time at 20% ultrasonication power; (**d**) *k* value of EG/PEID composite with different ultrasonication power sonication for 40 s; (**e**) comparison of thermal conductivity value of polymer composites in previous works.

**Figure 4 nanomaterials-12-01877-f004:**
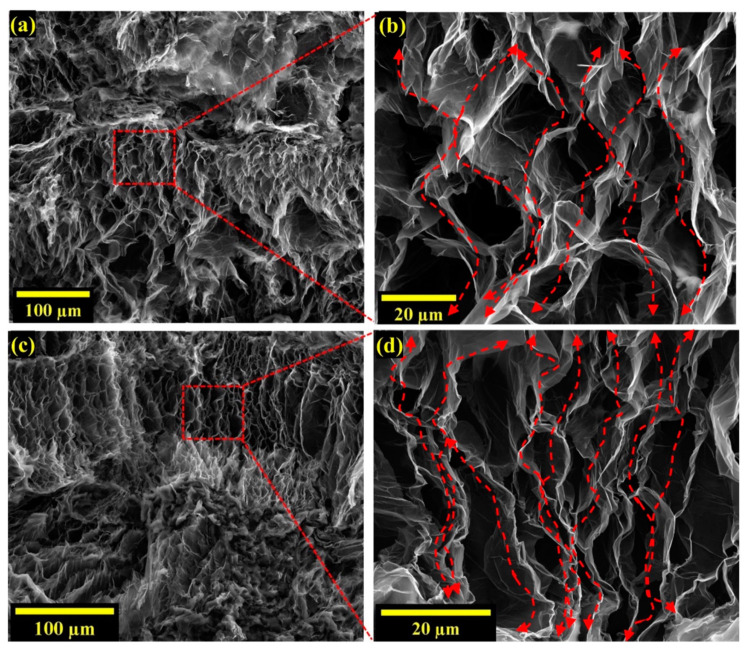
FE−ESEM images of EG/PEID composites with (**a**,**b**) 7.5 wt%, (**c**,**d**) 10 wt% fabricated at 20% amplitude for 40 s ultrasonication; (**a**,**c**) 350×magnification, (**b**,**d**) 3500× magnification.

**Figure 5 nanomaterials-12-01877-f005:**
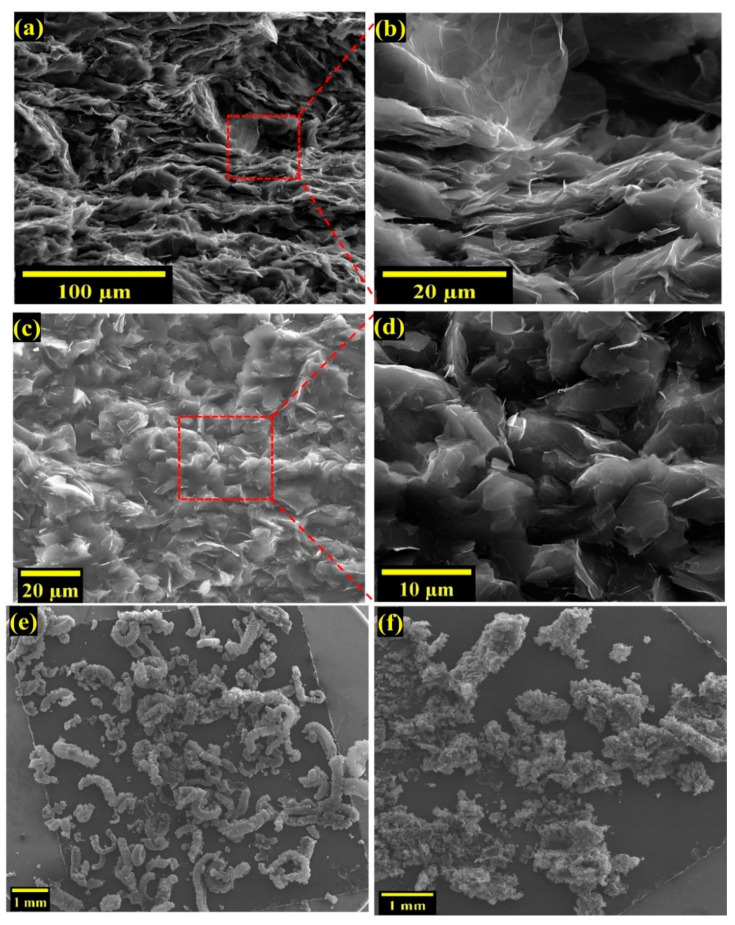
FE−ESEM images of (**a**,**b**) 10 wt% EG/PEID composite with ultrasonication for 3 min at 20% amplitude, (**c**,**d**) 10 wt% GnP/PEID composite with ultrasonication for 40 s at 20% amplitude; FE−ESEM images of EG at 20% sonication for (**e**) 40 s and (**f**) 3 min using montage large area mapping.

**Figure 6 nanomaterials-12-01877-f006:**
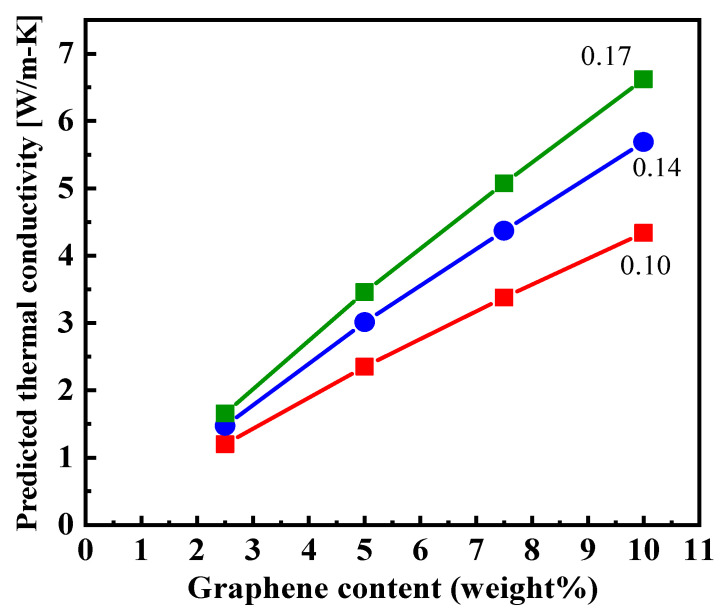
Predicted thermal conductivity value based on the effective medium theory (EMT) with different graphene–graphene interlayer thermal conductivity.

**Figure 7 nanomaterials-12-01877-f007:**
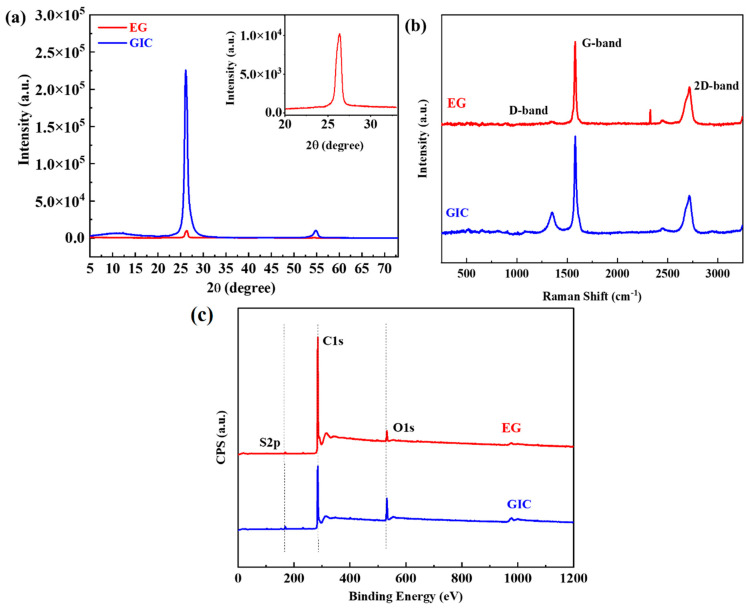
XRD spectra (**a**), RAMAN spectra (**b**), and XPS spectra (**c**) of GIC and EG filler.

**Table 2 nanomaterials-12-01877-t002:** The values of different parameters used in EMT calculations.

Material Parameters	Values
Average graphene lateral length, *l*,	10 μm
Average graphene thickness	10 nm
Aspect ratio of the graphene filler	0.001
Thermal conductivity of polymer phase	0.23 $W\cdot m^{-1}\cdot K^{-1}$
Thermal conductivity of graphene filler, $k_{1}$ and $k_{3}$ ($W\cdot m^{-1}\cdot K^{-1}$)	2000 and 10 $W\cdot m^{-1}\cdot K^{-1}$
Thermal conductivity of interlayer with Kapitza resistance	$0.04\;W\cdot m^{-1}\cdot K^{-1}$
Thermal conductivity of the interlayer with a firmly developed graphene–graphene contact state,	$0.17\;W\cdot m^{-1}\cdot K^{-1}$

**Table 3 nanomaterials-12-01877-t003:** Atomic composition by XPS analysis of EG and GIC.

	Atomic Composition by XPS (at%)		
	C (285.08 eV)	O (532.08 eV)	S (169.11 eV)
GIC	85.14	13.16	1.7
EG	95.76	4.24	-

## Data Availability

Any new data presented are available upon reasonable request by the authors.

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
