# Peer review of "Large Enhancement in Thermal Conductivity of Solvent−Cast Expanded Graphite/Polyetherimide Composites"

_nanomaterials, 2022, doi:10.3390/nano12111877_

Round 1
Reviewer 1 Report
This manuscript deals with the prepatation of polyetherimide-graphene and epoxy-graphene nanocomposites as potential candidates of materials with high thermoconductivity. Based on the importance of the topic the manuscript is acceptable for the publication after the Major revision of the following:
- There are some logically icorrect sentences in the text (line 56, line133).
- Figure 1a looks not very clear and needs improvement.
- For their experiments the authors used DMF (bp 153 C). At the temperature of 140 C and higher this solvent readily decarbonilates to form CO and HNMe2, and, possibly, these two gases may influence on the porosity of the resulting composite. I would suggest to the authors to carry out one blank experiment by using DMA (b.p. 165 C) and compare the results.
Reviewer 2 Report
The paper by Tarannum et al. concerns the preparation and characterization of the expanded-graphite/polyetherimide composites. Although the subject is important, the way of presentation is scientifically unacceptable. Critically, the conclusions are not supported by the results since the claimed covalent functionalization is not proven. The paper is very messy: missing upper indices in units e.g. 6.56 "Wm-1K-1" throughout the manuscript, wrongly written names of chemicals ("N-N, Dimethylformamide"), mixed up fonts, unacceptable quality of figures (e.g. fig1a, fig1g - Value on the scale unclear, Unclear labels of the scale bars in all (!) figures). Also, presenting the enhancements in thermal conductivity in % is misleading since the thermal conductivity of just matrices is missing in Table 1. Also, where does the sulfur come from in GIC? What is happening to other functional properties of the composites (mechanical, electrical, optical)? Not addressing the above areas of research significantly diminishes the impact of the work. To summarize, in the current form the manuscript cannot be published.
Reviewer 3 Report
The article : “Large enhancement in thermal conductivity of solvent cast expanded-graphite/polyetherimide composites” Fatema Tarannum et Al. reports on a very large enhancement in thermal conductivity of polyetherimide-graphene and epoxy-graphene nanocomposites prepared via solvent casting technique ascribed to expanded graphite..
The work is generally well written and reports a detailed study of the graphene nanocomposites however there are some elements to be discussed
Figure: Figure 1 a the writing in the figure seems to be incomplete and the figure is not described in detail in the text or in the caption. Please check,
Figure 2 : The chemical structure of PEID is of poor quality I suggest to redesign it.
Figure 3 : check carefully the letters (a), (b) ... they seem to be cut and some writings are difficult to read.
I suggest adding a figure schematizing the proposed mechanism for the large enhancement in thermal conductivity of graphene nanocomposites
Lines 56-58 Authors wrote: ” Intercalation of acid molecules and oxidizing agents into graphite (Fig. 1c) followed by rapid heating (900 ï‚°C) leads to a conversion of intercalation agents into a gaseous state (Fig. 1a) resulting in an expansion of graphite structure”. These features seem not to be schematized in figure 1. please check
What about the thermal conductivity of polyetherimide-graphene and epoxy-graphene nanocomposites prepared via solvent casting technique over the time? And what about thermal stability and thermal ageing of the composites?
Lines 133-139. Some reference are missing see below.
“Polyetherimide and epoxy, the polymers chosen for this work, also enable the advantage of allowing superior thermal interaction with graphene” add some references.
“…strong thermal interaction through hydrogen bonding with the oxygen groups in expanded-graphene…” add some references.
“…Evidence for presence of oxygen groups in expanded graphite is provided through X-ray photoelectron spectroscopy…” add some references.
In Experimental Details as sonication is an important element in the preparation of nanocomposites please add the details of the instrumentation.
I suggest adding thermal analysis before and after the addition of expanded graphite
References: number, distribution over the time and self-citations are adequate for the work
Round 2
Reviewer 3 Report
The manuscript has been improved and is now suitable for publication